# Tuna: Instruction Tuning using Feedback from Large Language Models

**Haoran Li[1,†], Yiran Liu[3,‡], Xingxing Zhang[2], Wei Lu[1], Furu Wei[2]**

[1]StatNLP Research Group, Singapore University of Technology and Design
[2]Microsoft Research Asia, [3]Tsinghua University
haoran2_li@mymail.sutd.edu.sg, wei_lu@sutd.edu.sg
liu-yr21@mails.tsinghua.edu.cn, {xizhang,fuwei}@microsoft.com

## Abstract

Instruction tuning of open-source large language models (LLMs) like LLaMA, using direct outputs from more powerful LLMs such as Instruct-GPT and GPT-4, has proven to be a cost-effective way to align model behaviors with human preferences. However, the instruction-tuned model has only seen one response per instruction, lacking the knowledge of potentially better responses. In this paper, we propose finetuning an instruction-tuned LLM using our novel *probabilistic ranking* and *contextual ranking* approaches to increase the likelihood of generating better responses. Probabilistic ranking enables the instruction-tuned model to inherit the relative rankings of high-quality and low-quality responses from the teacher LLM. On the other hand, learning with contextual ranking allows the model to refine its own response distribution using the contextual understanding ability of stronger LLMs. Furthermore, we apply probabilistic ranking and contextual ranking sequentially to the instruction-tuned LLM. The resulting model, which we call **Tuna**, consistently improves the performance on Super Natural Instructions (119 test tasks), LMentry (25 test tasks), Vicuna QA, and can even obtain better results than several strong reinforcement learning baselines. Our code and data are available at https://github.com/microsoft/LMOps.

## 1 Introduction

Large language models (LLMs) have made significant progress by scaling up model size and data size (Peters et al., 2018; Devlin et al., 2019; Radford et al., 2019; Brown et al., 2020; OpenAI, 2023) for unsupervised pre-training and subsequently applying reinforcement learning from human feedback (RLHF) to align model responses with human preferences (Christiano et al., 2017; Ouyang et al., 2022). More recently, instruction tuning (Wei et al.,

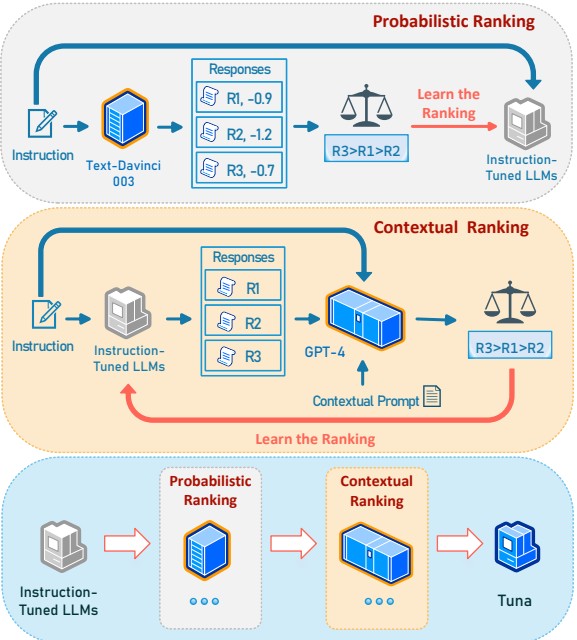

Figure 1: The finetuning process using probabilistic ranking (top), contextual ranking (middle), and a combination of both (bottom).

2022) with Self-Instruct algorithm (Wang et al., 2022a) has emerged as a cost-effective method for aligning with human preferences. In this approach, open LLMs like LLaMA (Touvron et al., 2023) can be finetuned on instruction-following data generated by OpenAI GPT using the Self-Instruct algorithm. The Alpaca model (Taori et al., 2023) exemplifies this technique, which enables close alignment with human preferences while reducing dependence on human-labeled data.

However, instruction tuning offers only a broad guideline for the base LLMs to transition from "next token prediction" to a more interactive, instruction-following style. As a result, the model may learn some superficial features or styles from the instruction data but still lacks a deeper understanding of what constitutes a preferred response. For instance, when given a question like "Give

---

†, ‡ Work done during internship at MSRA.

three tips for staying healthy", a base LLM may generate fluent yet undesirable continuations, while an instruction-tuned LLM could offer three general tips. Humans might prefer more detailed tips over general tips, but such tips are less likely to be sampled since they have lower likelihood within the current model distribution. This can be attributed to the fact that they are either unseen during instruction tuning or hard to be sampled due to the exposure bias (Ranzato et al., 2015).

To address this, we propose further finetuning of an instruction-tuned LLM to discern the quality of multiple responses more precisely, using our novel probabilistic ranking (Sec. 2.2; Fig. 1 top) and contextual ranking (Sec. 2.3; Fig. 1 middle) approaches. Probabilistic ranking enables the instruction-tuned LLM to inherit the high-quality and low-quality responses as well as their relative rankings from the teacher LLM (e.g., text-davinci-003). In contrast, contextual ranking aims to re-balance the instruction-tuned model's own response distribution with the help of stronger LLMs (e.g., GPT-4), mitigating the exposure bias issue.

We apply probabilistic ranking and contextual ranking sequentially to an instruction-tuned model, i.e., Alpaca (Taori et al., 2023), resulting in a model called **Tuna** (Sec. 2.4; Fig. 1 bottom). We evaluate Tuna on various benchmarks, including Super Natural Instructions (Wang et al., 2022b), which contains 119 diverse test tasks; LMentry (Efrat et al., 2022), comprising 25 tasks to assess the basic capabilities and robustness of LLMs; and Vicuna QA (Chiang et al., 2023) which evaluates the model's ability to answer a diverse set of questions with the assistance of GPT-4. Experimental results demonstrate that the Tuna model not only consistently outperforms the standard instruction-tuned models on all benchmarks, but also surpasses several strong RLHF baselines (Ouyang et al., 2022).

To summarize, our contributions are as follows:

- We propose *probabilistic ranking* and *contextual ranking*, which enable the instruction-tuned model to distinguish high-quality and low-quality responses and assign higher probability to the former accordingly.

- The **Tuna** model, obtained by sequentially applying probabilistic ranking and contextual ranking on an instruction-tuned LLM, achieves better results than several strong benchmarks, including RLHF models;

- Our model, data and code will be released to facilitate future research.

## 2 Methodology

In this section, we describe how to obtain our **Tuna** model using the feedback from LLMs. We first describe the vanilla instruction tuning. We then introduce our probabilistic ranking and contextual ranking approaches. Lastly, we describe how to integrate both ranking approaches.

### 2.1 Instruction Tuning

LLMs like GPT-3 (Brown et al., 2020) have been trained on a massive text corpus using maximum likelihood estimation (MLE):

$$L_{\text{MLE}}(y) = -\frac{1}{|y|} \sum_t \log p(y_t|y_{<t}; \theta), \quad (1)$$

where $\theta$ represents the parameters of the base model. The pre-training objective function compels the model to predict the next token $y_t$ given its prefix $y_{<t} = [y_0, y_1, ..., y_{t-1}]$. A sufficiently-trained LLM can generate fluent continuations given almost any prefix. However, the generated continuations may not align well with human preferences. As the primary goal of an LLM is to assist humans, it becomes essential to encourage the generation of content that follows human instructions and aligns with human preferences. The current dominant approach to enhance LLMs' instruction-following ability is called *instruction tuning* (Mishra et al., 2021; Wei et al., 2022; Taori et al., 2023), which finetunes the base LLMs in a supervised manner on instruction-response pairs $\{i, r\}$ (where $i$ is an instruction and $r$ is its response) using MLE:

$$L_{\text{MLE}}(i, r) = -\frac{1}{|r|} \log p(r|i; \theta'), \quad (2)$$

where $\theta'$ represents the parameters of the instruction-tuned model. After instruction tuning, we expect the model distribution $p(\cdot|i; \theta')$ to allocate higher probabilities to proper responses like $r$ rather than undesirable continuations.

Note that the responses in instruction-response pairs can either be annotated by humans[1] or generated by strong LLMs, such as Instruct-GPT or GPT-4 (Wang et al., 2022a). A prevalent and cost-effective approach for generating instruction tuning data is the Self-Instruct algorithm (Wang et al.,

---

[1]https://huggingface.co/datasets/databricks/databricks-dolly-15k

2022a). Specifically, it uses a strong LLM, e.g., text-davinci-003, to create instructions based on a few seed instructions, and then generates a single response for each instruction using the same LLM.

## 2.2 Probabilistic Ranking

Instruction tuning with the data generated by the Self-Instruct algorithm is essentially a form of sequence-level distillation (Kim and Rush, 2016). The rationale behind this class of distillation method is that the current commercial LLMs have significantly better capabilities than their open-source counterparts. Instead of learning from the single-response data, our *probabilistic ranking* approach leverages the relative rankings of multiple responses based on the teacher model's probabilities for better pseudo label distillation (see Fig. 1 top).

Let $r$ denote the original response for instruction $i$ in the instruction tuning dataset. We query strong (teacher) LLMs, such as text-davinci-003, to generate $N$ new responses for $i$. Let $r^{(0)}, r^{(1)}, \ldots, r^{(N-1)}$ denote these new responses, and $p(r^{(0)}|i), p(r^{(1)}|i), \ldots, p(r^{(N-1)}|i)$ denote their probabilities. While the teacher LLMs are expected to produce responses of comparable quality on average, there will inevitably be some variation in the quality of the generated responses. This inherent variability manifests itself in various aspects, such as differences in accuracy (Wang et al., 2023a), response length, and level of details provided (Wang et al., 2023b).

Intuitively, if a model is perfectly distilled, the relative probabilities assigned to two samples should be the same as those of the teacher model. Specifically, let $p(r^{(j)}|i; \theta')$ and $p(r^{(k)}|i; \theta')$ denote the probabilities of $r^{(j)}$ and $r^{(k)}$ w.r.t. the student model. If $p(r^{(j)}|i) > p(r^{(k)}|i)$, then $p(r^{(j)}|i; \theta') > p(r^{(k)}|i; \theta')$. We use the following normalized log-likelihood as the teacher model quality score to account for differences in response lengths:

$$s(i, r^{(k)}) = \frac{\log p(r^{(k)}|i)}{|r^{(k)}|^\beta}, \quad k = \{0, ..., N-1\}$$
(3)

where $|r^{(k)}|$ is the length of $r^{(k)}$ and $\beta$ represents the length penalty.

We then rank those responses in decreasing order based on $s(i, r^{(k)})$. The resulting instruction-response pairs become $\{i, r, (r^{[0]}, ... r^{[N-1]})\}$, where $i, r$ are from the original instruction tuning

data, and $r^{[j]}$ is considered to have better quality than $r^{[k]}$, if $j < k$. Once we obtain the ranked responses, we can encourage our model to learn from these rankings using a pairwise ranking objective, which has been successfully employed in previous work (Zhong et al., 2020; Liu et al., 2022; Zhang et al., 2022; Zhao et al., 2023). The ranking objective function is as follows:

$$L_{\text{rank}} = \sum_{0 \le j < k \le N-1} L_{\text{rank}}^{j,k}$$
(4)

$$L_{\text{rank}}^{j,k} = \max\left(0, v_{\theta'}^k - v_{\theta'}^j + m \times (k-j)\right), j < k$$
(5)

where $v_{\theta'}^k = \frac{1}{|r^{[k]}|} \log p\left(r^{[k]}|i; \theta'\right)$, $m > 0$ is the margin hyper-parameter. The ranking loss, $L_{\text{rank}}$, aims to teach the model to distinguish good responses from bad ones based on the teacher LLM's perspective. In addition to $L_{\text{rank}}$, we also apply a cross-entropy loss on the original response as regularization:

$$L = L_{\text{rank}} + \lambda L_{\text{MLE}}, \quad L_{\text{MLE}} = \frac{1}{|r|} \log p(r|i; \theta')$$
(6)

where $r$ is the original response, and $\lambda > 0$ controls the importance of $L_{\text{MLE}}$, which helps prevent over-optimization of the ranking loss.

After learning with probabilistic ranking, the model can better assign probabilities to superior and inferior responses.

## 2.3 Contextual Ranking

During the instruction tuning or the probabilistic ranking stage, the model is finetuned to generate a good $r$ given an instruction $i$. However, given the same $i$ during inference, the model may still generate a relatively low-quality response $r'$. This is related to the exposure bias problem (Ranzato et al., 2015), where the model fails to generate $r$ due to accumulated errors during the auto-regressive generation process. To address this issue, we use our *contextual ranking* approach to refine the distribution of responses generated by the model itself, assigning higher probabilities to better responses with the help of strong LLMs (Fig. 1 middle), thus alleviating exposure bias (Ranzato et al., 2015).

For each instruction, we first sample $N$ responses from the instruction-tuned model itself, i.e., $r^{(0)}, r^{(1)}, ..., r^{(N-1)} \sim p(\cdot|i; \theta')$. We hope the samples to be diverse enough so that better responses are more likely to appear in the sampled results.

To ensure diversity, we impose a constraint on the ROUGE-L (Lin, 2004) score between each pair of responses, requiring it to be less than a threshold $\tau$. If the ROUGE-L score exceeds $\tau$, we increase the sampling temperature and resample another response. If multiple trials still result in a ROUGE-L score above $\tau$, we retain the least similar response from the trials. After obtaining $N$ responses, we leverage the contextual understanding ability of commercial LLMs, such as GPT-4 (OpenAI, 2023), to rank them based on various aspects. The ranking process consists of multiple steps. First, we ask GPT-4 to assess whether the instruction requires an open-ended answer (e.g., story generation) or a close-ended answer (e.g., solving a math problem). We then request GPT-4 to generate its own response as a reference. Next, GPT-4 compares the reference response with the $N$ responses from different aspects and assign scores to each response. For open-ended instructions, GPT-4 evaluates relevance (score 0-5), level of details/justification (score 0-5), and accuracy (score 0-5) of the model responses compared to its reference response. For close-ended instructions, the evaluation criteria are accuracy (score 0-5), level of details/justification (score 0-5), and clarity (score 0-5). Finally, GPT-4 ranks responses in decreasing order based on the sum of their scores (see Appendix E for our complete prompt). We also manually evaluated GPT-4 rankings, which have achieved a strong correlation with human judgements (see Appendix G, H).

As in Sec. 2.2, the resulting instruction tuning dataset becomes $\{i, r, (r^{[0]}, ...r^{[N-1]})\}$. Note that the $r^{[k]}, 0 \leq k \leq N-1$, is derived from the instruction-tuned model itself. Lastly, we use the same objective function as in Eq. 6 to encourage the model to assign higher probabilities to better responses.

## 2.4 Integrating Probabilistic and Contextual Ranking

Given an instruction-tuned model, there are several options for further finetuning: 1) learning with probabilistic ranking alone; 2) learning with contextual ranking alone; 3) learning with probabilistic ranking followed by contextual ranking (see Fig. 1 bottom). We refer to the models finetuned with these three methods as **Tuna$_p$**, **Tuna$_c$**, and **Tuna**, respectively.

To optimally integrate both probabilistic ranking and contextual ranking techniques, it is recom-

mended to first obtain a Tuna$_p$ model, followed by applying contextual ranking to Tuna$_p$'s response distribution, resulting in the Tuna model. There are two reasons for this choice. First, although it is beneficial to learn the ranking of different responses from the teacher LLM's perspective (probabilistic ranking), the model might not fully capture the teacher's ranking knowledge due to its limited capacity. Second, contextual ranking enables the model to better adapt to its own capacity by working with the model's own generations. By generating its own responses, the model can finetune its understanding with the help of stronger LLMs and more effectively produce responses that are both closer to human preferences and compatible with its capacity constraints, alleviating the exposure bias issue (Ranzato et al., 2015).

## 3 Experiments

### 3.1 Model and Data

In our experiments, we use a 7B LLaMA model (Touvron et al., 2023) as the base model. The instruction tuning data is sourced from Alpaca (Taori et al., 2023), which consists of 52K instructions paired with responses that are generated by `text-davinci-003` using the Self-Instruct algorithm (Wang et al., 2022a). We perform instruction tuning on 52K Alpaca data using recommended hyperparameters, such as a learning rate of 2e-5 and the AdamW optimizer $(0.9, 0.999)$ (Loshchilov and Hutter, 2019).[2] For simplicity, we also refer to the instruction-tuned model as **Alpaca**.

For probabilistic ranking, we input 52K instructions from Alpaca dataset into `text-davinci-003` to produce $N = 4$ responses per instruction along with their log-likelihoods[3], with an inference temperature of 1. We calculate response scores using Eq. 3 with $\beta$ being 1.3, and rank the responses accordingly. Subsequently, we finetune the Alpaca model for 1 epoch with a learning rate 1e-5, margin $m = 0.1$, and cross entropy regularizer weight $\lambda = 1.0$. We denote the model trained exclusively with probabilistic ranking as **Tuna$_p$**.

For contextual ranking, we sample $N = 4$ responses from the Alpaca model with temperature $T = 1$ for each instruction. To avoid similar generations, we ensure the pairwise ROUGE-L (Lin, 2004) between responses is less than $\tau = 0.8$. Oth-

---

[2]https://github.com/AetherCortex/Llama-X
[3]GPT-4 is more powerful but it does not return log-likelihoods.

| | Super NI | | LMentry | Vicuna QA | | |
| | 0-shot | 2-shot | LMentry Score | Win | Lose | Tie |
|---|---|---|---|---|---|---|
| LLaMA | 11.0 | 23.6 | 26.3 | 4% | 92% | 4% |
| T5-LM 11B | - | 30.2 | 20.6 | - | - | - |
| T0 11B | - | 32.3 | 31.6 | - | - | - |
| InstructGPT 175B | - | 52.1 | 48.4 | - | - | - |
| Alpaca | 36.0 | 44.5 | 31.4 | - | - | - |
| + PPO-sim | 31.9 (-4.1) | 37.5 (-7.0) | 27.8 (-3.6) | 79% | 16% | 5% |
| + PPO-sim-GPT4-20K | 37.1 (+1.1) | 44.9 (+0.4) | 27.8 (-3.6) | 74% | 22% | 4% |
| $\text{Tuna}_p$ | **39.4 (+3.4)** | 43.9 (-0.6) | **35.0 (+3.6)** | 68% | 27% | 5% |
| $\text{Tuna}_c$ | 37.7 (+1.7) | **46.6 (+2.1)** | 32.2 (+0.8) | 74% | 20% | 6% |
| $\text{Tuna}_c$ (PRM) | 34.2 (-1.8) | 40.1 (-4.4) | 32.2 (+0.8) | 75% | 19% | 6% |
| Tuna | **38.7 (+2.7)** | **45.0 (+0.5)** | **34.7 (+3.3)** | **86%** | **10%** | **4%** |

Table 1: Performance comparison of different models on Super NI, LMentry and Vicuna QA. The numbers in bold indicate the top-2 results. The numbers in parentheses indicate the performance differences compared to Alpaca. The results of T5-LM 11B (Raffel et al., 2020), T0-11B (Sanh et al., 2022), InstructGPT 175B (Ouyang et al., 2022) are taken from Wang et al. (2022b); Efrat et al. (2022). The RLHF baselines PPO-sim and PPO-sim-GPT4-20K, which apply the PPO algorithm (Schulman et al., 2017), are taken from Dubois et al. (2023).

| Model | Alpaca | Alpaca+PPO-sim | $\text{Tuna}_p$ | $\text{Tuna}_c$ | Tuna |
|---|---|---|---|---|---|
| Score | 2.13 | 2.95* | 2.98* | 3.15* | 3.80*† |

Table 2: Human evaluation on Vicuna QA. * denotes that the model is significantly ($p < 0.01$) better than Alpaca, while † denotes that Tuna is significantly ($p < 0.01$) better than other models.

erwise, we remove the similar response, increase the temperature by 0.1, and resample. If three trials fail to produce unique enough responses, we keep the least similar one. We then employ GPT-4 to rank responses for the first **13K** instruction data with the GPT-4 inference temperature to be 0. The contextual ranking prompt is shown in Table 9.[4] The finetuning hyperprameters follow those of probabilistic ranking. We refer to the model trained on 13K contextual ranking data of the Alpaca model as **$\text{Tuna}_c$**.

Furthermore, we use the 13K GPT-4 ranking data to train a proxy ranking model (PRM) based on StableLM-3B.[5] The PRM is employed to re-rank Alpaca's responses on 52K instructions. We refer to the Alpaca model trained with 52K ranking data totally generated by the PRM as **$\text{Tuna}_c$ (PRM)**.

Lastly, we also collect 13K GPT-4 contextual ranking data based on $\text{Tuna}_p$'s responses instead of Alpaca's. We refer to the model finetuned on $\text{Tuna}_p$ as **Tuna**.

We also included strong reinforcement learning baselines for comparison (i.e., PPO-sim and PPO-sim-GPT4-20K models from AlpacaFarm (Dubois et al., 2023)).[6]

## 3.2 Evaluation

**Super Natural Instruction (Super NI)** Super NI (Wang et al., 2022b) contains 119 test tasks designed to evaluate a model's cross-task generalization ability. It includes a variety of classification and generation tasks, such as textual entailment and title generation. We report both 0-shot and 2-shot performance, where 0-shot provides only an instruction (referred to as "definition" in their literature) and 2-shot offers two additional positive examples. The evaluation metric for all 119 tasks is **ROUGE-L** (Lin, 2004), which is strongly correlated with human evaluation with a Pearson coefficient of 0.998 according to Wang et al. (2022b). Greedy decoding is applied during inference.

**LMentry** LMentry (Efrat et al., 2022) is a benchmark that primarily focuses on the accuracy and ro-

---

[4]The cost of calling OpenAI API is listed in Appendix B.
[5]https://github.com/Stability-AI/StableLM

[6]We also trained our own RLHF model, which is not as good as the ones in AlpacaFarm. The comparison can be found in Appendix I

bustness aspects of LLMs' generations. It contains 25 short tasks that are trivial to humans but challenging for LLMs. The final metric is **LMentry score**, which is calculated by multiplying its mean accuracy on 25 tasks with the robustness score. The model will be evaluated in a 0-shot manner, and greedy decoding is applied during inference.

**Vicuna QA** Vicuna QA (Chiang et al., 2023) comprises 80 test questions across 9 categories that measure an LLM's ability to generate relevant, detailed and accurate responses and it has been widely adopted in many works. Instead of having a ground truth for evaluation, it conducts pairwise comparisons with the help of GPT-4 (OpenAI, 2023). It prompts GPT-4 to compare the outputs of our models to the Alpaca model. We report the `win/lose/tie` rate against the Alpaca model.

**Human Evaluation** Additionally, we conduct human evaluations on Vicuna QA. Specifically, responses from five anonymous systems, namely Alpaca, Alpaca + PPO-sim, Tuna, $\text{Tuna}_p$, and $\text{Tuna}_c$, were randomly shuffled and presented to annotators who were then asked to rank these outputs. The scoring was designed such that the $i$-th ranked system receives a score of $6 - i$, meaning the best-ranked system receives a score of 5, and the worst-ranked system receives a score of 1. Each question was annotated by two different annotators, and the score was averaged.

### 3.3 Main Results

The main results are presented in Table 1. After instruction tuning, Alpaca demonstrates significant performance improvements over LLaMA on all three benchmarks. This highlights the successful transition from the "next token prediction" paradigm to a more interactive instruction-following paradigm.

Furthermore, both contextual and probabilistic ranking enhance performance across all three benchmarks. Specifically, $\text{Tuna}_c$ exhibits more improvement on the Super NI[7] 2-shot results while $\text{Tuna}_p$ performs better on Super NI 0-shot and LMentry, narrowing the performance gap with much larger models like InstructGPT-175B. Since the 2-shot input is longer than 0-shot, we conjecture that contextual ranking might be more beneficial for longer sequence generation than probabilis-

tic ranking. On the Vicuna QA benchmark, both $\text{Tuna}_p$ and $\text{Tuna}_c$ outperform Alpaca significantly on nearly 70% of the questions, as evaluated by GPT-4. Upon comparison with the RLHF baselines, $\text{Tuna}_p$ and $\text{Tuna}_c$ consistently demonstrate superior performances on both the Super NI and LMentry benchmarks. However, when it comes to the Vicuna QA benchmark, their performance is marginally lower than that of the RLHF baselines. Moreover, Tuna achieves the best performance on Vicuna QA while maintaining competitive scores on Super-NI and LMentry. Human results on Vicuna QA (see Table 2) also confirm that humans prefer the responses from our models.

Furthermore, $\text{Tuna}_c$ (PRM) demonstrates comparable performance to $\text{Tuna}_c$ on Vicuna QA and LMentry, but it underperforms both $\text{Tuna}_c$ and Alpaca on Super NI. This suggests that although the PRM has primarily learned ranking from the GPT-4 contextual ranking data, it also introduces some noise during the learning process. Overall, it is more effective to learn directly from GPT-4 contextual ranking data.[8]

### 3.4 Ablation Study

In this subsection, we delve deeper into the performance of our approach by examining several aspects, including: (a) the effect of more responses in instruction tuning, (b) the order of applying two ranking methods, (c) the influence of the cross entropy regularization, (d) the amount of probabilistic ranking data, and (e) the risks of GPT-4 evaluation.

**More Responses in Instruction Tuning** We explore whether Tuna's effectiveness is solely due to the increased response data by examining the impact of adding more responses per instruction during instruction tuning. We create a new model, Alpaca-Mul, by adding four extra responses from the probabilistic ranking dataset to the Alpaca dataset and fine-tuning the LLaMA model using Eq. 2. The results are presented in Table 3.

Upon evaluation on Super NI, Alpaca-Mul's performance is nearly identical to that of Alpaca but falls short when compared to the 0-shot settings of $\text{Tuna}_p$ and Tuna. On LMentry, Alpaca-Mul outperforms Alpaca, yet it still does not reach the performance levels of $\text{Tuna}_p$ and Tuna. Interestingly, in the Vicuna QA task, Alpaca-Mul slightly underperforms compared to Alpaca.

---

[7]ROUGE is used as the default metric on Super NI. However, our results follow the same trend using BERTScore (see Appendix J).

[8]Experiments with more PRMs can be found in App. D.

|  | Super NI | | LMentry | Vicuna QA | | |
| --- | --- | --- | --- | --- | --- | --- |
|  | 0-shot | 2-shot | LMentry Score | Win | Lose | Tie |
| Alpaca | 36.0 | 44.5 | 31.4 | - | - | - |
| Alpaca-Mul | 34.7 (-1.3) | **45.7 (+1.2)** | 33.9 (+2.5) | 42% | 53% | 5% |
| Tuna$_p$ | **39.4 (+3.4)** | 43.9 (-0.6) | **35.0 (+3.6)** | 68% | 27% | 5% |
| Tuna | **38.7 (+2.7)** | 45.0 (+0.5) | 34.7 (+3.3) | **86%** | **10%** | **4%** |
| Tuna$_c$ | 37.7 (+1.7) | **46.6 (+2.1)** | 32.2 (+0.8) | **74%** | **20%** | **6%** |
| Tuna$_{cp}$-13K | 35.7 (-0.3) | 44.0 (-0.5) | 33.5 (+2.1) | 58% | 37% | 5% |
| Tuna$_{cp}$-39K | 34.8 (-1.2) | 43.4 (-1.1) | **35.4 (+4.0)** | 46% | 48% | 6% |
| Tuna$_{cp}$-52K | 35.0 (-1.0) | 42.6 (-1.9) | 33.8 (+2.4) | 51% | 41% | 8% |
| mix-Tuna-52K | 37.7 (+1.7) | 44.2 (-0.3) | 30.0 (-1.4) | 70% | 23% | 7% |
| mix-Tuna-104K | 36.0 (+0.0) | 40.0 (-4.5) | 32.6 (+1.2) | 55% | 40% | 5% |

Table 3: Different combinations of probabilistic ranking data and contextual ranking data. The numbers in bold represent the top-2 results. The numbers in parentheses represent the performance difference compared to Alpaca.

These findings suggest that merely adding more responses without differentiating them does not necessarily lead to improved response generation. Overall, the results of Alpaca-Mul indicate that Tuna's superior performance cannot be solely attributed to the availability of more response data.

**Integration Order** An alternative approach to Tuna involves first training the Tuna$_c$ model, and subsequently continuing training the Tuna$_c$ model with probabilistic ranking data. The resulting model is referred to as Tuna$_{cp}$.

We explore various strategies for training Tuna$_{cp}$: 1) finetuning Tuna$_c$ with the first 13K probabilistic ranking data (Tuna$_{cp}$-13K); 2) finetuing Tuna$_c$ model with last 39K probabilistic ranking data (Tuna$_{cp}$-39K); 3) finetuning Tuna$_c$ model with 52K probabilistic ranking data (Tuna$_{cp}$-52K). Additionally, we also try to finetune original Alpaca model with a combination of 13K GPT-4 contextual ranking data (generated from Alpaca model's responses) and the last 39K probabilistic ranking data (mix-Tuna-52K). We also finetune Alpaca model with 52K contextual ranking data (13K GPT-4 contextual ranking + 39K ranking-model-generated data) plus 52K probabilistic ranking data (mix-Tuna-104K). The training details are listed in the Appendix C. The results are listed in Table 3.

None of the combination strategies consistently outperform both Tuna$_p$ and Tuna$_c$ across the Vicuna QA and Super NI benchmarks. On LMentry, however, finetuning Tuna$_c$ with probabilistic ranking data is beneficial, especially when no duplicate data is present (Tuna$_{cp}$-39K). This suggests

| rank | 1 | 2 | 3 | 4 |
| --- | --- | --- | --- | --- |
| contextual ranking | 66.4 | 55.2 | 51.4 | 44.8 |
| prob. ranking | 55.8 | 54.3 | 52.5 | 49.4 |
| PRM | 69.2 | 57.8 | 50.9 | 44.7 |

Table 4: The average ranking lengths of contextual ranking data, probabilistic ranking data and the data generated by the proxy ranking model (PRM).

that shorter probabilistic ranking data are beneficial when high accuracy and robustness are top priority.

Interestingly, Tuna$_{cp}$ is not comparable to Tuna, indicating that the order in which the model is trained with contextual and probabilistic ranking matters. One plausible explanation is that both the original Alpaca data and the probabilistic ranking data are generated by text-davinci-003, while Tuna$_c$ has significantly shifted the model distribution by re-ranking the Alpaca model's responses, making it challenging to finetune Tuna$_c$ with probabilistic ranking data again.

**The Effect of Cross Entropy Regularizer** We examine the influence of the weight $\lambda$ of the cross entropy regularizer in Eq. 6 on performance by varying $\lambda$ across different values: $\{0, 0.1, 1, 5, 10\}$ while training the Tuna$_c$ model. Fig. 2 illustrates that as $\lambda$ increases, the performance on accuracy-oriented benchmarks such as Super NI and LMentry improves, while the performance on open questions does not necessarily follow the same trend. On one hand, this finding suggests that with a small $\lambda$, learning with contextual ranking may induce

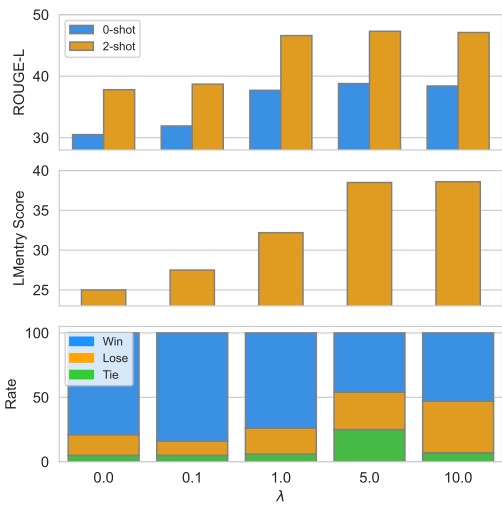 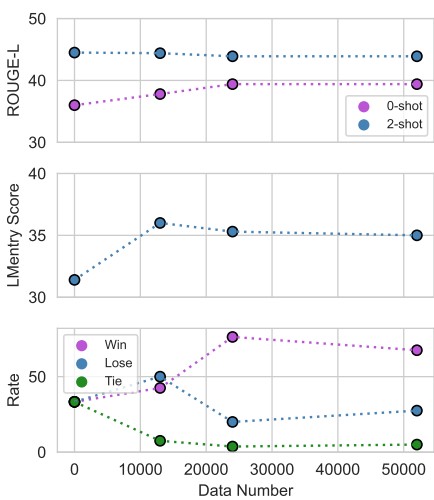

Figure 2: The effect of varying the weight $\lambda$ of cross entropy regularization in Eq. 6 on $\text{Tuna}_c$. The win/lose/tie rate on Vicuna is computed against Alpaca.

Figure 3: The effect of varying the number of probabilistic ranking data on $\text{Tuna}_p$.

long and detailed answers, but those answers are not always accurate. On the other hand, it implies that accuracy-oriented benchmarks and open QA benchmarks are complementary, and researchers should consider more diverse test cases to thoroughly evaluate a model (Wang et al., 2023b).

**The Amount of Probabilistic Ranking Data** We investigate the impact of varying the amount of probabilistic ranking data used for finetuning the $\text{Tuna}_p$ model by testing different data sizes, i.e., $\{0, 13000, 24000, 52000\}$. 0 refers to the Alpaca model. The results, shown in Fig. 3, reveal that for probabilistic ranking, 13K data points are sufficient for Super NI and LMentry, while Vicuna QA requires 24K data points. We conjecture that this saturation phenomenon can be attributed to two reasons. First, 52K Alpaca instructions generated by Self-Instruct algorithm are not diverse enough, as new instructions are produced by `text-davinci-003` using prompt instructions sampled from a limited seed task pool. Second, instruction tuning itself may only require a limited amount of data to perform behavior cloning, as discussed in Zhou et al. (2023). Thus, we can further reduce the cost of probabilistic ranking data generation by half.

**The Risks in GPT-4 Evaluation** We present evidence that evaluating a model on open QA with the help of GPT-4 may be risky. Table 4 displays the ranking length of our proxy ranking model (PRM). It shows that the PRM has inherited GPT-4 ranking's bias towards longer outputs (Li et al., 2023).

However, as we discussed in Sec. 3.3, the data generated by the PRM is not as good as the original 13K contextual ranking data, as assessed by more targeted automatic evaluations like Super NI and LMentry. Despite the inferior quality of the PRM-generated data, the performance on Vicuna QA remains almost unaffected (see $\text{Tuna}_c$ (PRM) in Table 1). This observation suggests that evaluating LLMs on open QA with GPT-4 may not always be as accurate as it appears, echoing the findings of Wang et al. (2023b). It highlights the need for more representative test questions or additional targeted benchmarks for evaluation.

## 4 Related Work

**Instruction Tuning** Instruction tuning aims to improve the usability of base language models (Brown et al., 2020; Raffel et al., 2020; Chowdhery et al., 2022) by finetuning them on instruction-response pairs in a zero-shot (Wei et al., 2022) or few-shot manner (Mishra et al., 2021; Wang et al., 2022b; Mallen et al., 2023). The instruction data can be sourced from off-the-shelf NLP benchmarks (Mishra et al., 2021; Wei et al., 2022; Wang et al., 2022b) or generated by LLMs (Wang et al., 2022a; Honovich et al., 2022; Taori et al., 2023; Peng et al., 2023).

**Ranking Loss** Learning through re-ranking sequence-level outputs has been studied in sequence-to-sequence models (Wiseman and Rush, 2016; Edunov et al., 2018; Liu et al., 2022; Zhang et al., 2022). BRIO and MoCa algorithms (Liu

et al., 2022; Zhang et al., 2022) adopt a pairwise ranking loss to guide the model to generate summaries with higher ROUGE scores (Lin, 2004). In this paper, we use GPT-4's (OpenAI, 2023) strong contextual understanding ability and `text-davinci-003`'s (Ouyang et al., 2022) intrinsic probability measures for ranking. In parallel with our work, Yuan et al. (2023) also propose pairwise ranking loss for finetuning LLMs. Key differences include: 1) our pipeline finetuning strategy; 2) our focus on ranking the model's responses; 3) our use of the original response for cross entropy regularization, while they select the highest-reward response. Additionally, Liu et al. (2023c) also employs GPT models for finetuning BART (Lewis et al., 2019) on the summarization task.

**Pre-Trained Model Evaluation** Large pretrained models are powerful evaluation metrics due to their strong contextual understanding ability, such as BERTScore (Zhang* et al., 2020), BARTScore (Yuan et al., 2021), MoverScore (Zhao et al., 2019), COMET (Rei et al., 2020), and GPTScore (Fu et al., 2023). More recently, there are more evaluation strategies based on GPT-3.5 and GPT-4 (Liu et al., 2023b; Gao et al., 2023).

## 5 Conclusion

In this paper, we propose to finetune an instruction-tuned LLM using our probabilistic ranking approach ($\text{Tuna}_p$), contextual ranking approach ($\text{Tuna}_c$), and a combination of both (Tuna). Our comprehensive experiments demonstrate consistent performance improvements across three benchmarks: Super Natural Instructions (119 test tasks), LMentry (25 test tasks), and vicuna QA. Furthermore, our methods outperform popular reinforcement learning from human feedback baselines that rely on the proximal policy optimization algorithm. These findings underscore the effectiveness of our approach in enhancing the performance of instruction-tuned LLMs and pave the way for future research in this area.

## Limitations

Despite the promising results achieved by our Tuna model, there are several limitations that should be acknowledged. The first limitation is GPT-4 ranking inconsistency. In our experiments, we relied on GPT-4 for contextual ranking, which may introduce bias due to the inconsistency in its ranking performance. As a powerful LLM, GPT-4 is generally expected to provide accurate and reliable rankings; however, it may still be sensitive to the phrasing or structure of prompts (Dubois et al., 2023). This inconsistency may lead to suboptimal rankings and potentially affect the overall performance of the Tuna model. In future work, it would be beneficial to design more robust prompts that can mitigate the impact of GPT-4's ranking inconsistencies. Another limitation is the evaluation benchmark. In this paper, we evaluated the Tuna model on three benchmarks, which provided a diverse range of tasks and challenges. However, it is unclear how well the Tuna model would generalize to other types of tasks, domains, or languages. Further research is needed to explore the applicability of the Tuna model to a broader range of problems and settings. The last limitation is the reliance on the use of proprietary LLMs, such as GPT-4 and `text-davinci-003`, for generating responses and rankings. This dependency may limit the accessibility and reproducibility of our method for researchers who do not have access to these proprietary models. Developing alternative methods that can leverage open-source LLMs or other ranking mechanisms would be a valuable direction for future research.

## Acknowledgements

We would like to thank reviewers for their valuable feedback. This research/project is supported by Ministry of Education, Singapore, under its Tier 3 Programme (The Award No.: MOET320200004), the National Research Foundation Singapore and DSO National Laboratories under the AI Singapore Program (AISG Award No: AISG2-RP-2020-016), and Ministry of Education, Singapore, under its Academic Research Fund (AcRF) Tier 2 Programme (MOE AcRF Tier 2 Award No: MOE-T2EP20122-0011). Any opinions, findings and conclusions or recommendations expressed in this material are those of the authors and do not reflect the views of the Ministry of Education, Singapore.

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

## A  The Length Penalty $\beta$ for Probabilistic Ranking Data

In our preliminary experiments, we found that the length penalty $\beta = 1.3$ is able to induce detailed responses and validated this choice on LIMA (Zhou et al., 2023) dataset. We finetune the $\beta$ parameter in Eq. 3 using the LIMA training dataset, which contains 1030 high-quality expert instruction annotations, allowing LLaMA-65B to be finetuned and achieve remarkably strong performance across a wide range of topics. Note that the training set also contains 50 modified Super NI examples but they are from the training tasks while we test our models on 119 Super NI test tasks. Specifically, we first obtain $\text{Tuna}_p$ models with probabilistic ranking data scored with different $\beta$. Then, we compute the token-level negative log-likelihood (NLL) of

the output of each LIMA instance under different $\text{Tuna}_p$ models and average the token likelihood over the whole LIMA training set. The results are shown in Table 5. It can be seen that with $\beta = 1.3$, the model can achieve the best NLL on LIMA training set. Thus, we set $\beta = 1.3$ in our experiments.

| $\beta$ | 0.9 | 1.0 | 1.1 | 1.2 | 1.3 | 1.4 |
|---|---|---|---|---|---|---|
| NLL | 2.14 | 2.12 | 2.11 | 2.10 | **2.09** | **2.09** |

Table 5: The token-level log-likelihood of LIMA training set under $\text{Tuna}_p$ models trained with probabilistic ranking data scored with different $\beta$.

## B  OpenAI API Pricing

We list the cost of calling OpenAI API models in Table 6.[9] The human labeling cost per 1K examples is estimated based on the pricing listed in Dubois et al. (2023), at 0.25\$ for each comparison. For each data example, there are 4 responses and thus $4 * (4 - 1)/2 = 6$ comparisons. Thus, the human labor cost per 1K examples is 1500\$.

## C  Training Details of $\text{Tuna}_{cp}$ and mix-Tuna

The hyperparameters are listed in Table 7. For models finetuned from Alpaca, i.e., $\text{Tuna}_c$, $\text{Tuna}_p$, and mix-Tuna, the learning rate is 1e-5. The only exception is mix-Tuna-104K, whose learning rate is 5e-6 since it contains 52K duplicate data. For models finetuned from $\text{Tuna}_c$ or $\text{Tuna}_p$, i.e., $\text{Tuna}_{cp}$ and Tuna, the learning rate is 1e-6. We use 8 Nvidia V100-32GB GPUs for all experiments in this paper.

## D  Other Proxy Ranking Models (PRM)

Similar to the PRM introduced in Sec. 3.1, we use the 13K GPT-4 ranking data to train another PRM based on LLaMA-7B, which we refer to as PRM-7B. We denote the PRM based on StableLM-3B as PRM-3B. These two ranking models are employed to re-rank Alpaca's responses on 52K instructions. The Alpaca model trained with 52K data totally generated by the ranking models are referred to as $\text{Tuna}_c$ (PRM-3B-52K) and $\text{Tuna}_c$ (PRM-7B-52K). Note that $\text{Tuna}_c$ (PRM-3B-52K) is the $\text{Tuna}_c$ (PRM) listed in Table 1. We denote the Alpaca model trained with 13K GPT-4 contextual ranking data plus the last 39K data generated by

---

[9] https://openai.com/pricing

|  | Data Num | Model | Price |
|---|---|---|---|
| Probabilistic Ranking | 52K | `text-davinci-003` | 275$ |
| Contextual Ranking | 13K | GPT-4-0314 | 380$ |
| Human | 1K | - | 1500$ |

Table 6: The estimated cost of calling OpenAI API and human labeling.

|  | LR | epoch | batch size | warmup |
|---|---|---|---|---|
| $\text{Tuna}_c$ | 1e-5 | 1 | 128 | 2 |
| $\text{Tuna}_p$ | 1e-5 | 1 | 128 | 2 |
| mix-Tuna-52K | 1e-5 | 1 | 128 | 2 |
| mix-Tuna-104K | 5e-6 | 1 | 128 | 2 |
| Tuna | 1e-6 | 1 | 128 | 2 |
| $\text{Tuna}_{cp}$-13K | 1e-6 | 1 | 128 | 2 |
| $\text{Tuna}_{cp}$-39K | 1e-6 | 1 | 128 | 2 |
| $\text{Tuna}_{cp}$-52K | 1e-6 | 1 | 128 | 2 |

Table 7: The hyperparameters of training different models.

the ranking models as $\text{Tuna}_c$ (PRM-3B-39K) and $\text{Tuna}_c$ (PRM-7B-39K).

The results are listed in Table 8. We can observe that models trained with ranking data generated by both PRMS do not achieve better results on Super NI compared to $\text{Tuna}_c$. The performances of $\text{Tuna}_c$ (PRM-3/7B-39K) is close to $\text{Tuna}_c$ (PRM-3/7B-52K), implying that the ranking model have learned 13K contextual ranking data well. Using a larger ranking model, such as 7B, does not gain better performance, which indicates that the ranking ability might not necessarily scale with the pre-training model's capacity. In general, the best strategy is still to learn directly from GPT-4 contextual ranking data, which contains less noise.

## E Contextual Ranking Prompt

We show the prompt that we use for GPT-4 contextual ranking in Table 9.

## F Is the Contextual Ranking Prompt Too Long?

In (Liu et al., 2023a), the authors found the "lost in the middle" phenomenon occurs at around 2K (20 documents * 100 tokens/document) tokens for GPT-4 (note we used GPT-4 in contextual ranking). We computed the average length of the prompt (including four responses and the ranking guidelines) used in GPT-4 ranking. The average length is 650 tokens, which is significantly shorter than 2K. Thus, the

input length does not seem to be an issue in GPT-4 ranking. Our human experiments above also confirm that the GPT-4 ranking is closely aligned with human assessments (see Appendix G).

## G Human Evaluation of GPT-4 Ranking

We conducted human evaluations of GPT-4 rankings on 50 questions used for contextual ranking. We asked annotators to rank the four system outputs produced by our model and we observe that the ranking quality by GPT-4 is reasonably good (the Spearman coefficient between the human rankings and GPT-4 rankings is 0.72). Furthermore, we also manually inspected the explanations given by GPT-4 for the ranking results. We found these explanations to be well-reasoned and plausible. Perhaps this is not surprising given the fact that several recent papers found GPT can be good evaluators in multiple NLP tasks (Fu et al., 2023; Gao et al., 2023). We believe the ranking feedback of this level is sufficient for guiding our model for better training (our experiments also proved this).

## H Why Not Choose Pairwise Ranking in GPT-4 Ranking

There are several reasons why ranking 4 responses together is preferred over pairwise rankings.

1. API cost: Pairwise ranking for four responses requires (4 * 3) / 2 = 6 API calls, significantly increasing the total cost. Moreover, a loop

| | Super NI | | LMentry | Vicuna QA | | |
| --- | --- | --- | --- | --- | --- | --- |
| | 0-shot | 2-shot | LMentry Score | Win | Lose | Tie |
| Alpaca | 36.0 | 44.5 | 31.4 | - | - | - |
| Tuna$_p$ | **39.4 (+3.4)** | 43.9 (-0.6) | **35.0 (+3.6)** | 68% | 27% | 5% |
| Tuna$_c$ | 37.7 (+1.7) | **46.6 (+2.1)** | 32.2 (+0.8) | 74% | 20% | 6% |
| Tuna$_c$ (PRM-3B-39K) | 35.6 (-0.4) | 40.4 (-4.1) | 33.4 (+2.0) | **79%** | **15%** | **6%** |
| Tuna$_c$ (PRM-7B-39K) | 33.5 (-2.5) | 40.3 (-4.2) | 32.5 (+1.1) | 73% | 20% | 7% |
| Tuna$_c$ (PRM-3B-52K) | 34.2 (-1.8) | 40.1 (-4.4) | 32.2 (+0.8) | 75% | 19% | 6% |
| Tuna$_c$ (PRM-7B-52K) | 34.6 (-1.4) | 41.1 (-3.4) | 32.0 (+0.6) | 73% | 20% | 7% |
| Tuna | **38.7 (+2.7)** | **45.0 (+0.5)** | **34.7 (+3.3)** | **86%** | **10%** | **4%** |

Table 8: Performance comparison of different models. The numbers in bold indicate the top-2 results. The numbers in parentheses indicate the performance difference compared to Alpaca.

(e.g., R1 > R2, R2 > R3, R3 > R1) could occur when R1/2/3 are of similar qualities, potentially requiring extra API calls for further validation.

2. The GPT-4 ranking quality is good enough, see Appendix G.

# I  Comparison between Our RLHF Models and PPO-sim/PPO-sim-GPT4-20K

We compare our RLHF models and PPO-sim/PPO-sim-GPT4-20K from Dubois et al. (2023) on Vicuna QA. The results can be found in Table 10. PPO-sim/PPO-sim-GPT4-20K have better responses and thus we choose to report the results of their models.

# J  BERTScore Results on Super NI

By default, ROUGE is employed on Super NI. We additionally reported BERTScore (which is proven to be a better alternative for ROUGE) in Table 11 and the results follow the same trends (also see Table 1).

# K  Examples

We show some examples in 12.

Below is an instruction that describes a task, paired with an input that provides further context. Write a response that appropriately completes the request.

### Instruction:
{Instruction}

### Input:
{Input}

### Response:

###Response 0:
{Response 0}

###Response 1:
{Response 1}

###Response 2:
{Response 2}

###Response 3:
{Response 3}

We would like you to rate Response 0/1/2/3 in reply to the given instruction displayed above.
First, identify if the instruction requires open-ended or close-ended responses.
Second, you need to generate one high quality '###Response 4' in answer to the instruction. It needs to have the same format as other responses and will be used as a reference later.
Third, identify if there are duplicate responses and keep only one of the duplicate responses for the following steps.
Fourth, compare Response 4 with Response 0/1/2/3/4 and assign each response an overall score on a scale of 0 to 15 where a higher score indicates better overall quality. For an open-ended instruction, please rate based on the relevance (score 0 to 5), level of details/justification: (score 0 to 5) and accuracy (score 0 to 5) of each response; for a close-ended instruction, please rate based on the accuracy (score 0 to 5), level of details/justification (score 0 to 5) and clarity (score 0 to 5) of each response. The ratings should have the format: 'Response k: [sum of the 3 individual scores you give to response k]'.
Last, rank the responses in decreasing order of their overall scores. The ranking should have the format: 'rank: [i, j ,k, l, m]'. If there are duplicate responses, keep only one of them in the rank, that is, the ranking may become: 'rank: [i, j, k, l]', 'rank: [i, j, k]' 'rank: [i, j]' or even 'rank: [i]'.

Table 9: Contextual Ranking Prompt for GPT-4.

|                              | Win  | Lose | Tie  |
| ---------------------------- | ---- | ---- | ---- |
| PPO-sim                      | 79%  | 16%  | 5%   |
| PPO-sim-GPT4-20K             | 74%  | 22%  | 4%   |
| Our PPO (w/ GPT4 ranking data) | 55%  | 35%  | 10%  |

Table 10: Vicuna QA results of our RLHF models and (Dubois et al., 2023).

| MODEL            | 0-shot    | 2-shot    |
| ---------------- | --------- | --------- |
| Alpaca           | 0.638     | 0.705     |
| PPO-sim          | 0.594     | 0.652     |
| PPO-sim-GPT4-20K | 0.641     | 0.714     |
| $Tuna_p$         | **0.667** | 0.690     |
| $Tuna_c$         | 0.655     | **0.727** |
| Tuna             | 0.665     | 0.699     |

Table 11: BERTScore evaluation of the model outputs in Super NI. Higher is better.

| Instruction: "Why is open-source important? Answer in 3-5 sentences" |
| --- |

**Alpaca**:
Open-source software is important because it allows for collaboration and innovation. It allows developers to work together to create better software, and it allows users to access the source code and make changes to the software to meet their own needs. Open-source software also encourages transparency and allows users to see how the software works and how it can be improved. Additionally, open-source software is often more secure and reliable than proprietary software, as it has been tested and reviewed by a large number of developers. Finally, open-source software is often free or low-cost, making it accessible to everyone.

**Tuna$_c$**:
Open-source software is important because it allows for collaboration and innovation. It allows developers to work together to improve existing software and create new software, and it allows users to view and modify the source code to customize the software to their own needs. Open-source software also encourages transparency and accountability, as anyone can view and inspect the code to ensure that it is secure and reliable. Additionally, open-source software is often more cost-effective than proprietary software, as users can access the source code and modify it to suit their own needs without having to pay for a license. Finally, open-source software is often more reliable and secure than proprietary software, as it is developed and tested by a large community of developers.

**Tuna$_p$**:
Open-source software is important because it allows developers to access the source code of a program and make changes to it. This allows developers to customize the software to their needs and to fix any bugs or errors that may be present. Additionally, open-source software is often free to use and modify, which makes it more accessible to those who may not be able to afford proprietary software. Open-source software also allows developers to collaborate and share ideas, which can lead to the development of better and more efficient software. Finally, open-source software can be used as a learning tool, as it allows developers to see how the software works and how it is put together. This can help them to better understand programming and coding in general.

**Tuna**:
Open-source software is important because it allows developers to collaborate on projects, share code, and create better products. By making the source code available to the public, developers can easily collaborate on projects, allowing them to create better products and services. This collaboration also allows developers to quickly identify and fix bugs, as well as add new features and functionality. Additionally, open-source software is often more secure than proprietary software, as it can be more easily audited and reviewed by the public. Finally, open-source software is often more cost-effective than proprietary software, as it does not require licensing fees or other costs associated with closed-source software. Open-source software is an important part of the software industry, and is used by many businesses and individuals.

Table 12: Example responses of different models.

Instruction: "Write a Python function that generates a random number between 1 and 10 (inclusive)."

Listing 1: Alpaca

```python
def random_number():
    return random.randint(1, 11)
```

Listing 2: Tuna$_c$

```python
def random_number():
    return random.randint(1, 11)
```

Listing 3: Tuna$_p$

```python
"""
Generate a random number between 1 and 10 (inclusive)
"""

import random

def random_number():
    return random.randint(1, 10)

if __name__ == '__main__':
    print(random_number())
```

Listing 4: Tuna

```python
"""
Generate a random number between 1 and 10 (inclusive)
"""

import random

def random_number():
    return random.randint(1, 10)

if __name__ == '__main__':
    print(random_number())
```

Table 13: Example responses of different models.