# OpenReview forum: "Tuna: Instruction Tuning using Feedback from Large Language Models"
_EMNLP/2023/Conference — EMNLP 2023 Findings_

### Official Review · Reviewer_ANTd · 2023-07-25

**Soundness:** 3

**Excitement:**

3: Ambivalent: It has merits (e.g., it reports state-of-the-art results, the idea is nice), but there are key weaknesses (e.g., it describes incremental work), and it can significantly benefit from another round of revision. However, I won't object to accepting it if my co-reviewers champion it.

**Missing References:**

Alex Mallen, Akari Asai, Victor Zhong, Rajarshi Das, Daniel Khashabi, and Hannaneh Hajishirzi. 2023. When Not to Trust Language Models: Investigating Effectiveness of Parametric and Non-Parametric Memories. In Proceedings of the 61st Annual Meeting of the Association for Computational Linguistics (Volume 1: Long Papers), pages 9802–9822, Toronto, Canada. Association for Computational Linguistics.

**Paper Topic And Main Contributions:**

The paper proposes a new framework for finetuning Instruction Tuning model (Student Model) with a feedback score from Large Language Models (LLMs) as Teacher Model. The framework consists of two ranking methods for diversifying the generation outputs of Student Model (smaller LLM).

The first method is referred to as Probabilistic Ranking, where Teacher Model (proprietary  GPT4) generates N-ranked diverse responses based log likelihood ranking. Then, Student Model uses the generated examples as ground truth to finetune the model on Instruction Tuning held-in data/task with new loss function: the aggregation of rank-based loss and weighted MLE Cross Entropy loss. The rank-based loss is defined as delta or the gap between the log likelihood of low rank responses versus high rank responses. The ranking loss is adopted from previous study (.., Zhao et al., 2023).

The second method is referred to as Contextual Ranking. That is, Student Model generates N-diverse responses by utilizing ROUGE-L diversity measure. Then, the generated responses are being evaluated by Teacher Model (GPT4). GPT4 is asked to rate the quality of reponses based on evaluation aspects (relevance, level of details, accuracy, etc) of two type of NLG objectives: Open-Ended and Close-Ended text generation.

The framework also explores the ablation and the ensemble of the two ranking methods as a continual process.

The main contribution of the paper is NLP Engineering experiment.

**Reasons To Accept:**

-	The experiments are reasonably executable, excluding the the utilization of propietary LLMs for researchers without API access.
-	The paper uses ablation study to validate the proposed method or framework, including varying different size of finetuning data and briefly discussing the risk of GPT4-based automatic evaluation.

**Reasons To Reject:**

My concerns of this manuscript are mainly two:

- The framework is highly dependent on propietary LLMs, which the authors have also mentioned in Limitation section.

The Student Model is forced to align with the preference of Teacher Model (GPT4), even though the risk of utilizing GPT4 as automatic evaluation metric is known, which authors have also discussed in the Results section. The risk, however, remains an open question. The pros is that the estimated cost for the API is less expensive than crowd-sourcing human. However, it does not necessarily mean that human is fully replaceable by larger LLMs. A feasible solution of those trade-offs is that authors can run human evaluation on few samples in which larger LLM performs differently (unfaithful vs faithful outcomes). For example, on samples where larger LLM has high performance (lower loss) versus samples where LLM has low performance score (higher loss).

- The model evaluation is rather unfair.

The authors compare between (i) models that are not aligned with GPT4 preference and (ii) aligned models (Student Models). So, intuitively the aligned models receive more support and can achieve a better performance results, particularly on held-out benchmark datasets in which GPT4 (Teacher Model) performs reasonably well. RLHF methods are taken directly from previous work’s results (Dubois et al., 2017), so it is not clear whether the resulting scores of PPO methods in Table 1 will be same if they are run on the same computing resource with the remaining models in the paper.

Side note:
An example of a more fair evaluation would be: comparing the performance between (i) models that is finetuned with human alignment ranking score vs. (ii) models that is finetuned with larger LLMs scoring. This human alignment setup might not be suitable for large scale finetuning data, so sampling on a smaller pool of data is a necessary precondition.

**Reproducibility:**

2: Would be hard pressed to reproduce the results. The contribution depends on data that are simply not available outside the author's institution or consortium; not enough details are provided.

**Reviewer Confidence:**

4: Quite sure. I tried to check the important points carefully. It's unlikely, though conceivable, that I missed something that should affect my ratings.

---

> ### Author Rebuttal · Authors · 2023-08-29
>
> Thanks for your insightful comments and constructive suggestions!
>
> ***Dependency on Proprietary LLMs***
>
> Our approach includes two distinct steps: probabilistic ranking and contextual ranking. It is important to clarify that this method does not necessitate the use of proprietary LLMs, despite our implementation utilizing OpenAI GPT models. Specifically, the probabilistic ranking step requires a teacher LLM capable of text completion and computing associated probabilities given a text prompt, whereas the contextual ranking step merely requires an LLM with text completion capabilities.
>
> We resorted to using proprietary LLMs in our experiments due to the substantial performance gap observed between open-source and proprietary LLMs at the time of our research. However, it is noteworthy that this disparity is progressively diminishing, as evidenced by the recent release of llama-2-70B-Chat in July 2023 and code-llama-34B in August 2023. It is anticipated that open-source LLMs comparable to GPT-3.5-turbo, at least in certain sub-domains, will be made available in the near future. These forthcoming models can be seamlessly integrated into our method.
>
> We acknowledge the importance of this discussion and will include it in the revised version of our manuscript.
>
> ***On Utilizing GPT-4 as Automatic Evaluation Metric***
>
> Thanks for analyzing the pros and cons of utilizing GPT-4 as automatic evaluation as well as kindly recommending a feasible solution on human evaluation! We are grateful for this. We agree that human is not fully replaceable by larger LLMs. Therefore, we added two human experiments.
>
> *1. Human Evaluation on Vicuna QA*
>
> We conducted a human evaluation experiment on the Vicuna QA test set, which consists of 80 questions, during the author response period. In this experiment, responses from five anonymous systems, namely Alpaca, Alpaca + PPO-sim, Tuna, Tuna_p, and Tuna_c, were randomly shuffled and presented to annotators who were then asked to rank these outputs. The scoring was designed such that the i-th ranked system receives a score of $6 - i$, meaning the best-ranked system receives a score of 5, and the worst-ranked system receives a score of 1. Each question was annotated by two different annotators. The table below presents the average human scores of the five systems:
>
> |      | Alpaca  | Alpaca + PPO-sim | Tuna_p | Tuna_c | Tuna |
> |------|:---:|:---:| :---: | :---: | :---:|
> | **Score** | $2.13$  | $2.95^{*}$  | $2.98^{*}$  | $3.15^{*}$  | $3.80^{\dagger *}$ |
>
> $*$ denotes that the model is significantly (p<0.01) better than Alpaca while $\dagger$ denotes that Tuna is significantly (p<0.01) better than other systems.
>
> Furthermore, we conducted a student-t test between Tuna and all the other systems, and the results indicate that Tuna is significantly better than the other systems in comparison (p < 0.01). We found that the rankings of these five systems are consistent with the GPT-4 evaluation in our paper, except for *Alpaca+PPO-sim* (in GPT-4 evaluation, PPO-sim is slightly better than Tuna_p and Tuna_c; while in human evaluation PPO-sim is slightly worse).
>
> *2. Human Evaluation on GPT-4 rank in contextual ranking*
>
> We conducted human evaluations of GPT-4 rankings on 50 questions used for contextual ranking. We asked annotators to rank the four system outputs produced by our model and we observe that the ranking quality by GPT-4 is reasonably good (the Spearman coefficient between the human rankings and GPT-4 rankings is 0.72). Furthermore, we also manually inspected the explanations given by GPT-4 for the ranking results. We found these explanations to be well-reasoned and plausible. Perhaps this is not surprising given the fact that several recent papers found GPT can be good evaluators in multiple NLP tasks [1][2]. We believe the ranking feedback of this level is sufficient for guiding our model for better training.
>
> We will add these results and explanations to the revised version of our manuscript.
>
>
>
> ***Unfair Model Evaluation***
>
> Actually we spent a significant amount of time to implement our own PPO (using the same GPT4 ranking data as Tuna_c to train the reward model). However, the performance is worse than both PPO-sim and PPO-sim-GPT4-20K from the public available AlpacaFarm [3](https://crfm.stanford.edu/2023/05/22/alpaca-farm.html). See the results on Vicuna QA below. We are convinced that the PPO-sim and PPO-sim-GPT4-20K implementations from AlpacaFarm are good. We therefore decide to run their models on the three benchmarks and report these results in the Table 1.
>
> |                                | Win | Lose | Tie |
> |--------------------------------|-----|------|-----|
> | PPO-sim                        | 79% | 16%  | 5%  |
> | PPO-sim-GPT4-20K               | 74% | 22%  | 4%  |
> | Our PPO (w/ GPT4 ranking data) | 55% | 35%  | 10% |
>
> Although AlpacaFarm PPOs and our PPO all leverage the Alpaca dataset for training, our PPO implmentation is worse perhaps because of the following reasons.
>
> 1) PPO-sim leverages annotations from multiple LLMs (i.e., GPT4, GPT3.5-turbo and text-davinci-003) to train their reward models.
> 2) AlpacaFarm used separate splits for the three stages (i.e., sft, reward modeling training and RL) of PPO.
> 3) Perhaps better hyper-parameter tuning. There are many hyper-parameters in PPO and four models need to be trained in PPO.
>
> Thanks again for the insightful comments and we will add these numbers and discussions above to our revised manuscript.
>
> ***Missing Reference***
>
> We will cite Mallen et al. (2023).
>
> ***Low Reproducibility***
>
> Last but not the least, we plan to release our models, code and data (i.e., GPT annotations from both Probabilistic Ranking and Contextual Ranking) as stated in our original manuscript (line 29 ~ 30) to help other researchers to reproduce our work.
>
> [1]. Jinlan Fu et al. GPTScore: Evaluate as You Desire. 2023.\
> [2]. Mingqi Gao et al. Human-like Summarization Evaluation with ChatGPT. 2023.\
> [3]. Yann Dubois et al. AlpacaFarm: A Simulation Framework for Methods that Learn from Human Feedback. 2023.

---

### Official Review · Reviewer_eUe6 · 2023-08-05

**Typos Grammar Style And Presentation Improvements:** 1. Space could be obtained by i) maki…
**Soundness:** 3

**Excitement:**

3: Ambivalent: It has merits (e.g., it reports state-of-the-art results, the idea is nice), but there are key weaknesses (e.g., it describes incremental work), and it can significantly benefit from another round of revision. However, I won't object to accepting it if my co-reviewers champion it.

**Paper Topic And Main Contributions:**

The paper proposed LLM fine-tuning approach using probabilistic and contextual ranking approaches to increase the likelihood of generating better responses. The authors addressed the open-source LLM finetuning issues using a powerful LLM output and claim the proposed fine-tuned model named “Tuna” perform well for many NLP tasks.

**Questions For The Authors:**

1. What is the basis of choosing the powerful LLM, can be explained clearly ?. Does it based on the LLM size, performance?
2. Is any human evaluation made to verify the ranking generated by the LLM ?.
3. What is the motivation to choose Alpca 52K instruction set for fine-tuning ?. Can explain this in the paper.
4. Apart from Rouge, metrics like BertScore could be considered for computing semantic similarity.


**Reasons To Accept:**


1. The proposed approach will be helpful in improving open-source LLM fine-tuning.
2. The fine-tuned model “Tuna” performed better as compared with state-of-the-art LLMs (LLaMA, T5-LM, InstructGPT) as illustrated in the paper.


**Reasons To Reject:**

The performance of the fine-tuned model depends on the teacher LLM used for fine-tuning and its not clearly defined in the paper about the performance of the model on complex tasks like answering critical reasoning and solving arithmetic problems.

**Reproducibility:**

4: Could mostly reproduce the results, but there may be some variation because of sample variance or minor variations in their interpretation of the protocol or method.

**Reviewer Confidence:**

5: Positive that my evaluation is correct. I read the paper very carefully and I am very familiar with related work.

---

> ### Author Rebuttal · Authors · 2023-08-29
>
> Thanks for your thorough review and thoughtful suggestions!
>
> ***On Choosing Powerful LLM***
>
> We choose the powerful LLMs mainly based on their performance. However, the performance of a LLM is strongly correlated with its size (i.e., larger models generally generalize better than smaller models)[1][2].
>
> We choose text-davinci-003 in probabilistic ranking, since it is the most powerful LLM with probability output; while we choose the most powerful LLM (i.e., gpt-4) in contextual ranking.
>
> These explanations above will be added to our updated manuscript.
>
>
> ***Human Evaluation on Ranking Generated by the LLM***
>
> We conducted human evaluations of GPT-4 rankings on 50 questions used for contextual ranking. We asked annotators to rank the four system outputs produced by our model and we observe that the ranking quality by GPT-4 is reasonably good (the Spearman coefficient between the human rankings and GPT-4 rankings is 0.72). Furthermore, we also manually inspected the explanations given by GPT-4 for the ranking results. We found these explanations to be well-reasoned and plausible. Perhaps this is not surprising given the fact that several recent papers found GPT can be good evaluators in multiple NLP tasks [3][4]. We believe the ranking feedback of this level is sufficient for guiding our model for better training (our experiments also proved this).
>
> We will add these results and analysis to our updated version.
>
>
> ***Motivation to Choose Alpaca 52K***
>
> We choose Alpaca 52K for several reasons.
> First, the Alpaca dataset contains a diverse set of instruction-response pairs and is widely used in open-source and research community. Second, it is free and publicly available, which facilitates reproducible research. Third, the size of this dataset is reasonably large and we observe the outputs of the resulting models look robust.
>
> We will add these explanations to our updated version.
>
>
> ***BERTScore Results***
>
> Here are the BERTScore and the original ROUGE-L results (also see Table 1 in our paper) on Super Natural Instruction (Super NI). The BERTScore results are consistent with the ROUGE-L results.
> | MODEL            | 0-shot     | 2-shot    |  0-shot ROUGE-L | 2-shot ROUGE-L |
> |--------------------|--------------|-------| --- | --- |
> | Alpaca           | 0.638      | 0.705     | 36.0 | 44.5 |
> | PPO-sim          | 0.594      | 0.652     | 31.9 | 37.5 |
> | PPO-sim-GPT4-20K | 0.641      | 0.714     | 37.1 | 44.9 |
> | Tuna_p           | **0.667**  | 0.690     | **39.4** | 43.9 |
> | Tuna_c           | 0.655      | **0.727** | 37.7 | **46.6** |
> | Tuna             | 0.665      | 0.699     | 38.7 | 45.0 |
>
> Note that we use `microsoft/deberta-xlarge-mnli_L40_no-idf_version=0.3.12` as the base model, which is recommended by the BERTScore authors.
>
> We will add these results above to our updated manuscript.
>
>
> ***Presentation Improvements***
>
> Thank you for your suggestions on improving the presentation of our paper, which is very valuable and practical for us. We will make the suggested changes, such as reducing the size of Tables 1 and 2, and streamlining the general explanation of instruction tuning and probabilistic ranking, to save space and enhance readability.
>
>
> ***Evaluation on Critical Reasoning and Solving Arithmetic Problems***
>
> According to [5], "there exists a very complex balance/tradeoff between language models’ multi-dimensional abilities; by paying the price of decreased generic ability, we can clearly lift up the scaling curve of models smaller than 10B towards a specialized multi-step math reasoning ability".
>
> We may need training data containing a significant amount of complex reasoning examples to increase the reasoning capabilities of models around 7B. Unfortunately, Alpaca 52K only contains a few such examples (it is more general) and reasoning specific dataset for instruction tuning does not seem to be ready. We therefore plan to leave the training of reasoning specific models in future work.
>
>
> [1]. Alon Brutzkus, Amir Globerson. Why do Larger Models Generalize Better? A Theoretical Perspective via the XOR Problem. ICML 2019.\
> [2]. Jared Kaplan et al. Scaling Laws for Neural Language Models. 2020.\
> [3]. Jinlan Fu et al. GPTScore: Evaluate as You Desire. 2023.\
> [4]. Mingqi Gao et al. Human-like Summarization Evaluation with ChatGPT. 2023.\
> [5]. Yao Fu et al. Specializing Smaller Language Models towards Multi-Step Reasoning. ICML 2023.

---

### Official Review · Reviewer_Eb1g · 2023-08-05

**Soundness:** 4

**Excitement:**

4: Strong: This paper deepens the understanding of some phenomenon or lowers the barriers to an existing research direction.

**Missing References:**

- This paper is similar to the [behavior cloning and RL fine-tuning](https://proceedings.neurips.cc/paper_files/paper/1996/file/68d13cf26c4b4f4f932e3eff990093ba-Paper.pdf) pipeline, in the robotics community.
The result (Tuna_{pc} is better than Tuna_{cp}) also proves that it's better to learn from the demonstration first, and then learn from the model exploration by itself.

**Paper Topic And Main Contributions:**

The paper proposes methods to improve the instruction-following abilities of Alpaca that have been instruction-tuned on data Self-Instruct.

The two proposed methods are: 1) Probabilistic ranking: Using ranking loss to finetune the LLM to learn relative rankings of responses from a teacher LLM like GPT-3, to distinguish high and low quality responses. Output logits from teacher model are needed in this step. 2) Contextual ranking: Finetune the LLM to rebalance its own response distribution using rankings from a stronger LLM like GPT-4, to assign higher probability to better responses rated by GPT-4.
The two methods are applied sequentially to get a model called Tuna. Experiments on 3 benchmarks - Super NI, LMentry, Vicuna QA - show Tuna outperforms the base instruction-tuned LLaMA model Alpaca. The author also showed that the order of Probabilistic ranking -> Contextual ranking is optimal, other orders or combinations are not able to get better performance. Tuna also outperforms Alpaca tuned with PPO.

**Questions For The Authors:**

A. In probablistic ranking, why using GPT-4 to rank 4 responses at the same time? considering that LLMs may [ignore the long context in the middle](https://arxiv.org/abs/2307.03172), it may be better to keep using pairwise comparison when prompting GPT-4. Need more analysis on this choice.

**Reasons To Accept:**

- Novel methods to improve on standard instruction tuning approaches using ranking losses.
- Thorough experiments validate improvements over strong baselines on multiple benchmarks.
- Good ablation study to show that the pipeline design is necessary and optimal.
- Well-written paper, easy to understand.

**Reasons To Reject:**

- There is no human evaluation of the quality of the model outputs. The benchmarks rely on automatic metrics like ROUGE-L or GPT-4 rating which may not fully capture quality. Human judgments could reveal if the improvements translate to better responses.
- In the probabilistic ranking approach, the authors have GPT-4 provide a single ranking for 4 generated responses to each instruction. However, [a recent paper](https://arxiv.org/abs/2307.03172) indicates LLMs tend to ignore context in the middle of prompts. The authors provide no analysis on why ranking 4 together was preferred over pairwise rankings.

**Reproducibility:**

4: Could mostly reproduce the results, but there may be some variation because of sample variance or minor variations in their interpretation of the protocol or method.

**Reviewer Confidence:**

4: Quite sure. I tried to check the important points carefully. It's unlikely, though conceivable, that I missed something that should affect my ratings.

**Typos Grammar Style And Presentation Improvements:**

- Captions of Table 2: raking -> ranking

---

> ### Author Rebuttal · Authors · 2023-08-29
>
> Thank you for your review and constructive feedback!
>
>
> ***Human Evaluation***
>
> We appreciate your insightful comment regarding the limitations of automatic metrics like ROUGE-L or GPT-4 rating in fully capturing the quality of model outputs. In response to this, we conducted a human evaluation experiment on the Vicuna QA test set, which consists of 80 questions, during the author response period. In this experiment, responses from five anonymous systems, namely Alpaca, Alpaca + PPO-sim, Tuna, Tuna_p, and Tuna_c, were randomly shuffled and presented to annotators who were then asked to rank these outputs. The scoring was designed such that the i-th ranked system receives a score of $6 - i$, meaning the best-ranked system receives a score of 5, and the worst-ranked system receives a score of 1. Each question was annotated by two different annotators. The table below presents the average human scores of the five systems:
>
> |      | Alpaca  | Alpaca + PPO-sim | Tuna_p | Tuna_c | Tuna |
> |------|:---:|:---:| :---: | :---: | :---:|
> | **Score** | $2.13$  | $2.95^{*}$  | $2.98^{*}$  | $3.15^{*}$  | $3.80^{\dagger *}$ |
>
> $*$ denotes that the model is significantly (p<0.01) better than Alpaca while $\dagger$ denotes that Tuna is significantly (p<0.01) better than other systems.
>
> Furthermore, we conducted a student-t test between Tuna and all the other systems, and the results indicate that Tuna is significantly better than the other systems in comparison (p < 0.01). We found that the rankings of these five systems are consistent with the GPT-4 evaluation in our paper, except for *Alpaca+PPO-sim* (in GPT-4 evaluation, PPO-sim is slightly better than Tuna_p and Tuna_c; while in human evaluation PPO-sim is slightly worse).
>
> We are grateful for your suggestion to incorporate human experiments into our evaluation. These results will be included in the revised version of our manuscript.
>
>
>
> ***Ranking Four Responses together v.s. Pairwise Rankings***
>
> There are several reasons why ranking 4 responses together is preferred over pairwise rankings.
>
> - API cost: Pairwise ranking for four responses requires (4 * 3) / 2 = 6 API calls, significantly increasing the total cost. Moreover, a loop (e.g., R1 > R2, R2 > R3, R3 > R1) could occur when R1/2/3 are of similar qualities, potentially requiring extra API calls for further validation.
>
> - We conducted human evaluations of GPT-4 rankings on 50 questions used for contextual ranking. We asked annotators to rank the four system outputs produced by our model and we observed that the ranking quality by GPT-4 is reasonably good (the Spearman coefficient between the human rankings and GPT-4 rankings is 0.72). Furthermore, we also manually inspected the explanations given by GPT-4 for the ranking results. We found these explanations to be well-reasoned and plausible. Perhaps this is not surprising given the fact that several recent papers found GPT can be good evaluators in multiple NLP tasks [1][2]. We believe the ranking feedback of this level is sufficient for guiding our model for better training (our experiments also proved this).
>
> We will add the explanations above to our updated version.
>
>
>
> ***Long Context (Lost in the Middle)***
>
> In [5], the authors found the "lost in the middle" phenomenon occurs at around 2K (20 documents * 100 tokens/document) tokens for GPT-4 (note we used GPT-4 in contextual ranking). We computed the average length of the prompt (including four responses and the ranking guidelines) used in GPT-4 ranking. The average length is 650 tokens, which is significantly shorter than 2K. Thus, the input length does not seem to be an issue in GPT-4 ranking. Our human experiments above also confirm that the GPT-4 ranking is closely aligned with human assessments.
>
> These disscusions will be added to the updated version of our paper.
>
>
>
> ***Missing References and Typos***
>
> We appreciate your suggestion to include references from the RL community related to behavior cloning and RL fine-tuning pipelines. We will add relevant citations [3][4] in our updated script. Additionally, we acknowledge the typos in Table 2 and are grateful for your attention to those details. These will be corrected in the updated manuscript.
>
>
>
> [1]. Jinlan Fu et al. GPTScore: Evaluate as You Desire. 2023.\
> [2]. Mingqi Gao et al. Human-like Summarization Evaluation with ChatGPT. 2023.\
> [3]. Stefan Schaal. Learning From Demonstration. NIPS 1996.\
> [4]. Faraz Torabi et al. Behavioral Cloning from Observation. IJCAI 2018.\
> [5]. Nelson F. Liu et al. Lost in the Middle: How Language Models Use Long Contexts. 2023

---

### Meta-Review · Area_Chair_FB6W · 2023-09-18

**Recommendation:** 3

**Metareview:**

The paper proposes methods to improve the instruction-following capabilities of the Alpaca model (learned through distilling from OpenAI models via self-instruct). The proposed method includes probabilistic ranking and contextual ranking that are sequentially applied to finetuning LLMs. The proposed method is evaluated on Super NI, LMentry, and Vicuna QA datasets and demonstrate its effectiveness.

---

### Decision · Program_Chairs · 2023-10-07

**Decision:**

Accept-Findings

**Comment:**

The paper proposes methods to improve the instruction-following capabilities of the Alpaca model (learned through distilling from OpenAI models via self-instruct). The proposed method includes probabilistic ranking and contextual ranking that are sequentially applied to finetuning LLMs. The proposed method is evaluated on Super NI, LMentry, and Vicuna QA datasets and demonstrate its effectiveness.